# AMPK Activation Downregulates TXNIP, Rab5, and Rab7 Within Minutes, Thereby Inhibiting the Endocytosis-Mediated Entry of Human Pathogenic Viruses

**DOI:** 10.3390/cells14050334

**Published:** 2025-02-24

**Authors:** Viktoria Diesendorf, Veronica La Rocca, Michelle Teutsch, Haisam Alattar, Helena Obernolte, Kornelia Kenst, Jens Seibel, Philipp Wörsdörfer, Katherina Sewald, Maria Steinke, Sibylle Schneider-Schaulies, Manfred B. Lutz, Jochen Bodem

**Affiliations:** 1Institute of Virology and Immunobiology, University of Würzburg, Versbacher Str. 7, 97078 Würzburg, Germany; viktoria.diesendorf@uni-wuerzburg.de (V.D.); veronica.larocca@fht.org (V.L.R.); michelle.teutsch@stud-mail.uni-wuerzburg.de (M.T.); haisam.alattar@stud-mail.uni-wuerzburg.de (H.A.); s-s-s@vim.uni-wuerzburg.de (S.S.-S.); manfred.lutz@uni-wuerzburg.de (M.B.L.); 2Fondazione Human Technopole–Viale Rita Levi-Montalcini, 1–Area MIND, 20157 Milano, Italy; 3Department of Microbiology and Immunology, Faculty of Pharmacy, Assiut University, Assiut 71515, Egypt; 4Fraunhofer Institute for Toxicology and Experimental Medicine ITEM, Biomedical Research in Endstage and Obstructive Lung Disease (BREATH), 30625 Hannover, Germany; helena.obernolte@item.fraunhofer.de (H.O.); katherina.sewald@item.fraunhofer.de (K.S.); 5Institute of Anatomy and Cell Biology, Koellikerstraße 6, 97070 Würzburg, Germany; kornelia.kenst@uni-wuerzburg.de (K.K.); philipp.woersdoerfer@uni-wuerzburg.de (P.W.); 6Bayer Vital GmbH, Medical Affairs Consumer Health Analgesics, Cough & Cold, 51368 Leverkusen, Germany; jenspeter.seibel@bayer.com; 7Fraunhofer Institute for Silicate Research ISC, Röntgenring 12, 97070 Würzburg, Germany; maria.steinke@uni-wuerzburg.de; 8Department of Oto-Rhino-Laryngology, Plastic, Aesthetic and Reconstructive Head and Neck Surgery, University Hospital Würzburg, Josef-Schneider-Straße 11, 97080 Würzburg, Germany

**Keywords:** acetylsalicylic acid, salicylic acid, antiviral activity, AMPK, ULK1, eEF2K, eEF2, receptor-mediated endocytosis, RNA viruses, PI3K/mTORC2 pathways, neuronal organoids, precision-cut lung slices

## Abstract

Cellular metabolism must adapt rapidly to environmental alterations and adjust nutrient uptake. Low glucose availability activates the AMP-dependent kinase (AMPK) pathway. We demonstrate that activation of AMPK or the downstream Unc-51-like autophagy-activating kinase (ULK1) inhibits receptor-mediated endocytosis. Beyond limiting dextran uptake, this activation prevents endocytic uptake of human pathogenic enveloped and non-enveloped, positive- and negative-stranded RNA viruses, such as yellow fever, dengue, tick-borne encephalitis, chikungunya, polio, rubella, rabies lyssavirus, and SARS-CoV-2, not only in mammalian and insect cells but also in precision-cut lung slices and neuronal organoids. ULK1 activation inhibited enveloped viruses but not EV71. However, receptor presentation at the cytoplasmic membrane remained unaffected, indicating that receptor binding was unchanged, while later stages of endocytosis were targeted via two distinct pathways. Drug-induced activation of the AMPK pathway reduced early endocytic factor TXNIP by suppressing translation. In contrast, the amounts of Rab5 and the late endosomal marker Rab7 decreased due to translation inactivation and ULK1-dependent proteasome activation within minutes. Furthermore, activation of AMPK hindered the late replication steps of SARS-CoV-2 by reducing viral RNAs and proteins and the endo-lysosomal markers LAMP1 and GRP78, suggesting a reduction in early and late endosomes and lysosomes. Inhibition of the PI3K and mTORC2 pathways, which sense amino acid and growth factor availability, promotes AMPK activity and blocks viral entry. Our results indicate that AMPK and ULK1 emerge as restriction factors of cellular endocytosis, impeding the receptor-mediated endocytic entry of enveloped and non-enveloped RNA viruses.

## 1. Introduction

Cellular metabolism must be tightly and rapidly adjusted to the availability of nutrients and growth factors. In response to alterations in the surrounding environment, the cell activates and inhibits kinase signalling pathways. Elevated levels of amino acids and growth factors trigger the activation of the phosphatidylinositol 3-kinase (PI3K)/Akt pathway, which exerts a positive regulatory effect on transcription by RNA polymerases I and II, as well as the translation of 5′ terminal oligopyrimidine tract (TOP) mRNAs. Additionally, the mammalian target of rapamycin (mTOR), a downstream kinase, regulates lysosomal and endoplasmic reticulum functions while inhibiting degenerative processes like lysosome acidification and autophagy.

Conversely, a glucose undersupply that results in reduced cellular ATP levels is sensed by the liver serine–threonine kinase B1 (LKB1), activating the AMP-activated protein kinase (AMPK). AMPK is a heterotrimeric complex that acts as the core energy sensor of cells (reviewed in [1]). AMPK phosphorylates the Unc-51-like autophagy-activating kinase (ULK1), thereby promoting autophagy. Simultaneously, the mTORC1 complex is inactivated through raptor phosphorylation, which results in lysosomal acidification due to the relief of suppression of V-ATPase by binding to the mTORC1 complex. Furthermore, AMPK has been shown to downregulate translation through phosphorylation of the eEF2 kinase, which in turn inactivates the translation elongation factor 2 (eEF2). eEF2 is essential for ribosome translocation. On the other hand, the p65 subunit of the NF-kappa B (NF-κB) transcription factor has been shown to reduce eEF2 kinase mRNA levels, thereby stimulating translation [2].

As suggested by prior research, the AMPK pathway may regulate endocytosis-dependent glucose uptake [1]. In addition, aspirin (acetylsalicylic acid (ASA)) has been shown to activate AMPK [3,4], which, in turn, inhibits vitamin C uptake in mice [5]. Furthermore, active AMPK negatively regulates glucose transporter 1 (GLUT1) endocytosis under low ATP levels to ensure glucose resupply. Under physiological conditions and in response to high glucose availability, thioredoxin-interacting protein (TXNIP) [6], a member of the α-arrestin family, hinders glucose uptake by binding to GLUT1 and promoting its internalisation, ubiquitination, and delivery to the lysosome [7,8]. AMPK phosphorylates TXNIP at serine residue 308, leading to its rapid degradation and thus enhancing glucose uptake by inhibiting the recycling of GLUT1. Similarly, Akt phosphorylates TXNIP at the same serine residue to ensure glucose uptake under conditions of high amino acid and growth factor supply [9].

TXNIP is also regulated by miR-183, which downregulates its expression. High nutrient availability activates the p65 subunit of NF-κB through the Akt/mTORC1/2 kinase pathway, thereby upregulating TXNIP via enhanced expression of histone deacetylase 2 (HDAC2) [10,11]. Thus, the Akt/mTORC1/2 kinase pathway alleviates the TXNIP-mediated suppression of GLUT1 endocytosis through Akt and NF-κB p65 [11].

Organelles in the endocytic system are defined by their Rab GTPases, which serve as platforms for the specific factors of the organelles [12]. Rab5 is associated with the outer membrane of the early endosome (EE) and is involved in endosome fusion [13]. Rab4 and Rab11 control the recycling of endosomes to the plasma membrane. Chemical activation of AMPK enhances Rab4 promoter activity [14]. Rab5 is activated by its guanine nucleotide exchange factor Rabex5, which binds to ubiquitinylated cargo. Rab5 recruits antigen 1 (EEA1), a tethering protein required for the fusion of endocytic vesicles with EEs, and the hexameric CORVET tethering complex, which functions in the fusion of endocytic vesicles and EEs [15,16]. CORVET binds to Rab5 and participates in the fusion of endocytic vesicles and EEs. During the switch from Rab5 to Rab7, Rab5 and Rabex5 are released from the endosome, and the Rab5-specific GTP-hydrolysis-activating protein inactivates the remaining Rab5 [17]. The expression of constitutively inactive Rab7 resulted in cargo enrichment in the EEs [18], probably by blocking the endosomal pathway. However, Rab7 is essential for transport from late endosomes (LEs) to lysosomes, where Rab7 and its downstream effector Rab-interacting lysosomal protein recruit the V-ATPase and control the stability of the V1G1-vATPase subunit [19]. Eventually, the LE fuses with the lysosome, releasing its content into the lysosomal lumen.

Recently, it has been reported that the Rift Valley fever virus (RVFV), an animal pathogen, is inhibited by AMPK activation. It has been concluded that an early step in the viral replication cycle is inhibited by AMPK, suggesting that RVFV infection is restricted due to the inhibition of fatty acid biosynthesis [20]. Previously, it has been shown that ASA and its metabolite salicylic acid (SA) inhibit SARS-CoV-2, influenza A, and rhinovirus replication in cell cultures and human precision-cut lung slices (PCLSs) [21,22,23]. In a more extensive retrospective cohort study, SARS-CoV-2 patients receiving ASA required less oxygen on admission than those not receiving ASA [24]. Here, we investigate how signalling cascades related to nutrient availability and potential AMPK activation by ASA and SA impact the replication of flaviviruses, including yellow fever (YFV), dengue (DENV), tick-borne encephalitis virus (TBEV), the alphavirus chikungunya, the coronavirus SARS-CoV-2, and rabies lyssavirus. Endemic areas of these viruses continue to expand. DENV is now endemic in over 100 countries across WHO regions, with the number of cases increasing from 505,430 in 2000 to 5.2 million in 2019 globally, with 2.8 million cases reported in the Americas alone in 2022. While an attenuated vaccine for DENV has been approved, YFV still causes 200,000 diagnosed cases and 30,000 deaths annually despite the introduction of the vaccine in 1937. Similarly, despite the approved rabies vaccines and postexposure prophylaxis, between 50,000 and 70,000 fatal canine rabies infections occur every year [25]. These findings indicate that the development of antiviral therapies remains an essential health issue.

It has been observed that SA efficiently activates AMPK through direct interaction [4]. While our previous work demonstrated the inhibitory effect of ASA/SA on SARS-CoV-2, the underlying mechanism remained unexplored. If the inhibition of SARS-CoV-2 by ASA/SA is indeed mediated through AMPK activation, it follows that the direct activation of AMPK might also inhibit other viruses. In the present work, we investigated whether an AMPK activator could similarly suppress the replication of different viruses, despite the differences in their replication cycles. We provide evidence that AMPK activation efficiently blocks the replication of multiple human pathogenic RNA viruses.

## 2. Materials and Methods

### 2.1. Viruses, Antibodies, Plasmids, and Chemicals

YFV strain: STAMARIL 17D-204; DENV type 2 strain [26]. The patient-derived SARS-CoV-2 and the chikungunya isolate have been described previously [27,28]. Wild-type rabies lyssavirus (015V-03645) was obtained from the FLI (Greifswald, Germany); the recombinant vaccine strain SPBN GAS was sourced from IDT Biologika (Dessau-Roßlau, Germany). The patient-derived rubella, enterovirus 71, and TBEV isolates were obtained from the virus diagnostics facility in Würzburg, Germany, and used with permission. Antibodies: Anti-PVR (CD155) Antibody, clone 4B3, ZooMab^®^ Rabbit Monoclonal recombinant, Sigma Aldrich; anti-LDL Receptor Antibody, clone 2N19, ZooMab^®^ Rabbit Monoclonal recombinant, Sigma Aldrich (Taufkirchen, Germany); anti-Rab5A Antibody #2143, Cell Signaling; anti-Rab7 (D95F2) XP^®^ Rabbit mAb #9367, Cell Signaling; and anti-TXNIP Antibody, clone 1K14 ZooMAb^®^ Rabbit Monoclonal, Sigma Aldrich; anti-β-Actin Antibody, clone 6L12, ZooMAb^®^ Rabbit Monoclonal, Sigma Aldrich; anti-SARS-CoV-2 Spike Protein (S1-NTD) Antibody #56996, Cell Signaling, anti-SARS-CoV/SARS-CoV-2 NSP8 Monoclonal Antibody (5A10), Invitrogen; and anti-LAMP1 (D4O1S) Mouse mAb #15665, Cell Signaling; anti-GRP78 Polyclonal Antibody, Invitrogen. All chemicals and inhibitors were purchased from Sigma-Merck (Darmstadt, Germany), MedChemExpress (Monmouth Junction, NJ, USA), and Carl Roth (Karlsruhe, Germany). Activators: AMPK (CAS 849727-81-7) and ULK1 (CAS 2101517-69-3).

### 2.2. Western Blotting Analysis

For Western blot analysis, cells were washed with ice-cold PBS buffer and lysed in RIPA buffer (150 mM NaCl; 1% NP40; 12 mM sodium-deoxycholate, 3.5 mM SDS; 50 mM Tris pH 7) with sonication. Then, Laemmli buffer (2% SDS, 10% glycerol, 60 mM Tris, 0.01% (*w*/*v*) bromophenol blue, 37.5 μM β-mercaptoethanol) was added. Proteins were separated by SDS PAGE (8–12%) and transferred to a nitrocellulose membrane using a wet blotting system (Carl Roth, Karlsruhe, Germany) and Towbin buffer (0.025 M Tris 0.192 M glycine, 20% methanol) for 1 to 3 h. Membranes were blocked with 5% non-fat milk in PBS buffer for 20 min and incubated with the respective antibodies overnight.

### 2.3. Cellular Proliferation Assays

Direct automatic cell counting determined cell proliferation, both with and without the compounds. Cells were seeded in optical plates (CellCarrier-96, PerkinElmer, Rodgau, Germany) and counted prior to the experiments. Subsequently, the compounds were added in decreasing concentrations, and the cells were incubated for three days. The cell numbers per well were determined using the PerkinElmer Ensight reader. Only those compound concentrations that did not significantly reduce the cell numbers per well were employed for antiviral assays.

### 2.4. Viral Infection, RNA Isolation, Detection, and Viral Load Determination

The cells were incubated with the compounds either 2 h before or 2 h after infection and then infected with the respective viruses. The medium was exchanged to remove inactive viruses, thereby influencing genome copy determination with the medium containing the compounds (except for AMPK or ULK1 activators). All infection experiments were performed in triplicate assays and repeated at least twice in independent experiments. After 48 or 72 h, 200 µL of the medium was collected, and viral genomes were purified using the High Pure Viral Nucleic Acid Kit (Roche, Mannheim, Germany). Genome quantification of DENV, SARS-CoV-2, and TBEV was performed with the respective LightMix assay (TIBMolBiol, Roche, Mannheim, Germany), while YFV assays were conducted as described previously [29]. The provided standard was used for genome copy-number quantification using the LightCycler 480 II V1.5 or Roche LightCycler 96 software V1.2 (Roche, Mannheim, Germany).

### 2.5. Generation of Neural Organoids and PCLS

Neural organoids were generated from human-induced pluripotent stem cells (hiPSCs), as previously described in detail [30]. PCLSs were prepared and infected as previously described [23,28,31].

### 2.6. FACS Staining and Analysis

After 2 h of treatment with AMPK or ULK, cells were washed with FACS buffer, and 1 × 10^6^ cells were resuspended in 50 µL of PBS supplemented with the primary antibodies and the viability dye on ice for 30 min. After that, the cells were washed with FACS buffer and counterstained with the corresponding conjugated secondary antibodies for 30 min on ice. The primary antibodies used were rabbit monoclonal anti-CD155 (1:100 dilution, clone: 4B3, Sigma-Aldrich, Taufkirchen, Germany) and mouse monoclonal anti-LDL receptor (1:100 dilution, clone: 2H7.1, Sigma-Aldrich, Taufkirchen, Germany). For live/dead staining, Fixable Viability Dye eFluor™ 780 was used (1:1000 dilution, cat. number: 65-0865-14, eBioscience, Life Technologies GmbH, Darmstadt, Germany). Goat anti-rabbit Alexa Fluor 647 (1:100 dilution, polyclonal, cat. number: A-21244, Invitrogen, Life Technologies GmbH, Darmstadt, Germany) and goat anti-mouse PE (1:100 dilution, polyclonal, cat. number: 550589, BD bioscience, Heidelberg, Germany) were used as secondary antibodies. Samples were measured using an Attune NxT Flow Cytometer and the FlowJo 10.10.0 software. FACS gating strategies can be found in the Appendix A.

### 2.7. Proteasome Activity Assay

The proteasome activity was assayed using the Proteasome-Glo™ 3-Substrate System (Promega, Walldorf, Germany). The cellular chymotrypsin-like, trypsin-like, and caspase-like protease activities were determined through luminescence assays, as described by the manufacturer. All assays were performed in six parallel reactions.

## 3. Results

### 3.1. AMPK Activation Suppresses Receptor-Mediated Dextran Uptake

Since it has been shown that GLUT1/4 endocytosis and RVFV entry are, although unrelated, inhibited by AMPK activation, we hypothesised that AMPK activation impedes, more broadly, receptor-mediated endocytosis. Thus, we sought to investigate whether AMPK activation inhibits the endocytosis of fluorescently labelled dextran, a marker of receptor-mediated endocytosis. After the dextran was taken up, the Huh-7 cells were incubated with SA or an AMPK activator ((CAS 849727-81-7), 69-3), and dextran was added to the medium for 2 h. The cells were fixed and subsequently analysed using confocal microscopy (Figure 1A). Indeed, the activation of AMPK by both compounds inhibited dextran uptake, indicating that the suppression of receptor-mediated endocytosis has occurred.

### 3.2. Activation of AMPK Inhibits Flaviviruses at Viral Entry

Since other (+)-stranded RNA viruses commonly utilise replicative compartments derived from endosomes, the endoplasmic reticulum, and lysosomes, we analysed whether AMPK activation by ASA/SA inhibits DENV, TBEV, and YFV replication. Huh-7 (DENV, YFV) or human glioma NCE U373 (TBEV) cells were pre-treated with 3 mM ASA/SA, 100 µM AMPK, or ULK1 activators for 2 h prior to infection (Figure 2A). The AMPK activator does not directly interact with AMPK but inhibits the mitochondrial complex I, reducing cellular ATP levels and subsequently activating AMPK [32]. The viruses were removed 2 h after infection through a medium exchange. Cell culture supernatants were collected after 2 (YFV, TBEV) or 3 (DENV) days; viral RNAs were isolated and quantified by RTqPCR (Figure 2C–F). All infection experiments detailed in this manuscript were performed three times in triplicate. The accumulation of TBEV and DENV RNAs was inhibited by more than 2 orders of magnitude (log_10_), demonstrating that ASA/SA treatment efficiently suppressed these viruses. Treatment with the AMPK activator inhibited TBEV replication by approximately 1.3 log_10_ and YFV by more than 3 log_10_, indicating that AMPK activation was sufficient to repress viral replication (Figure 2E,I,J). The activation of the downstream ULK1 further suppressed YFV and TBEV replication, showing that the inhibition of viral replication was mediated through ULK1.

Metformin, a commonly used antidiabetic drug, has also been shown to activate AMPK [33,34]. Given that high doses of SA are necessary for AMPK activation, we aimed to investigate whether a similar activation of AMPK could be achieved through the combination of metformin and SA. We anticipated that both compounds would exhibit additive effects. Cells were treated with increasing concentrations of SA, ranging from 0.185 to 3 mM, and metformin, from 0.625 to 10 mM, and were infected with YFV (Figure 2G). The EC_50_ for SA was calculated as 0.42 mM (0.62 mM on Vero cells) for AMPK activator 17.32 µM (Vero cells: 10.34 µM), while that for metformin was 1.91 mM (Vero cells: 1.61 mM). Concentrations of 1.5 mM SA suppressed YFV replication by 97.49%, whereas 10 mM metformin inhibited viral replication by 81.4%. The analysis of drug synergism using the SynergyFinder software version 3.0 [35] demonstrated that the compounds acted additively, leading to an inhibition of over 98% with 2.5 mM metformin and 1.5 mM SA, or with 10 mM metformin and 0.75 mM SA. Consequently, the combination of metformin and SA results in either enhanced inhibition of viral replication or requires lower concentrations of the compounds, thereby achieving comparable suppression. This suggests that the decreased concentrations of SA could lead to adverse effects on fever in patients.

Recently, we and others have observed that even direct antivirals can act in a cell-type-specific manner [31,36,37]. Consequently, we aimed to validate our findings regarding TBEV using patient-relevant systems, specifically employing human stem cell-derived 3D neuronal organoid models. The organoids were exposed to 3 mM SA or 100 mM ULK1 activator and were subsequently infected with TBEV. Cell culture supernatants were collected 2 days after infection, and viral RNAs were quantified by RTqPCR (Figure 2J). SA downregulated TBEV replication by more than 2 log_10_, and the ULK1 inhibitor reduced it by 6-fold. These results indicate that the activation of AMPK inhibits TBEV replication even in this model patient-near system. Since mosquitoes transmit YFV, we investigated whether AMPK activation influences viral replication in insect SF9 cells. We observed the antiviral effects of ASA and SA in these cells, providing evidence that the mechanism is conserved (Figure 2H).

Next, we aimed to determine which replication step was targeted by AMPK activation. Drugs were added either 2 h (Figure 2A) before (pre-treatment) or 2 h after viral infection (Figure 2B) to discern whether the compounds inhibited viral replication at the entry or the post-entry steps. For the latter setting, 3 mM ASA, SA and 100 µM AMPK activator (Figure 2D–I), or 100 µM of the ULK1, were used (Figure 2E,I). The cells were incubated for 46 h (YFV, TBEV) or 70 h (DENV). Viral RNAs were isolated and quantified by RTqPCR (Figure 2). We expected a lack of inhibition with AMPK activators, similar to the untreated controls, and wanted to determine whether AMPK activation affected only viral entry since the compounds were added after infection. We observed inhibition of viral replication by more than 2 log_10_ when compounds were added 2 h before viral entry. However, treatment with the compounds after infection did not reduce DENV infection, indicating that compound-mediated AMPK activation interfered with DENV entry. Experiments with YFV and TBEV yielded comparable results, suggesting that AMPK activation similarly inhibits viral entry. Remarkably, the pharmacological inhibition of receptor-mediated viral entry by the ULK1 activator suggested that this downstream kinase also regulates flavivirus entry, revealing a previously unrecognised role in the viral infection process.

We generated fluorescently labelled YFV to determine, using an independent assay, that entry is inhibited by AMPK activation. Cells were infected with concentrated YFV (MOI~10). After 1 day, the medium was exchanged for medium containing ATTO 643 DOPE, a compound that stains cell membranes. Cell culture supernatants were harvested after 12 h, and viruses were pelleted by ultracentrifugation, resulting in a coloured pellet. Huh-7 cells were treated with SA for 2 h; then, the medium was replaced with a medium containing a cytoplasmic membrane cell stain (BioTracker 400 blue, Sigma-Aldrich, Taufkirchen, Germany) but no compounds. This dye is taken up by endocytosis. The cells were infected with the stained viruses for 10 or 20 min and fixed with paraformaldehyde. Confocal microscopy revealed that YFV could not be detected in cells treated with SA for 10 min, indicating that the viruses were not endocytosed (Figure 2F). The membrane dye was not incorporated, suggesting a general block of endocytosis. However, after 20 min of infection without the compound, virus binding was restored, demonstrating that the inhibition was reversible. The results suggest that the AMPK activation blocks one of the first steps of endocytosis.

### 3.3. AMPK Activation Inhibits Enteroviruses, Alphavirus, and Rubella Virus Entry

Having observed that flavivirus entry is sensitive to AMPK activation, we sought to analyse whether this phenomenon also applies to other plus-stranded RNA viruses, such as rubella, chikungunya, poliovirus, and enterovirus 71. Like RVFV, the chikungunya virus enters the cell via the Mxra8 receptor. The myelin oligodendrocyte glycoprotein serves as a receptor for the rubella virus, while the scavenger receptor class B member 2, or the P-selectin glycoprotein ligand-1, functions as a receptor for enterovirus 71. Polioviruses utilise CD155 for receptor-mediated endocytosis [38,39,40,41]. Therefore, we expected that AMPK activation would inhibit early infection steps only if entry is regulated by cellular factors rather than by viral factors.

We compared viral replication in cells pre-treated for 2 h with the compounds or in cells infected 2 h prior to treatment to further analyse whether the activation of AMPK by SA or an AMPK activator suppresses viral entry into the cells. Viral replication was determined by RT-qPCR 2 d post-infection. Activation of AMPK suppressed chikungunya (SA: 2.14 log_10_, *p* = 1.4 × 10^−5^; AMPK activator: log_10_ = 0.9, *p* = 2.4 × 10^−5^) (Figure 3A), rubella (SA: log_10_ = 2.16, *p* = 0.03; AMPK activator: log_10_ = 2.82, *p* = 0.03) (Figure 3B), EV71 (SA: log_10_ = 2.16, *p* = 0.03; AMPK activator: log_10_ = 2.82, *p* = 0.002) (Figure 3C), and poliovirus (2.11 log_10_, *p* = 0.002) (Figure 3D) when cells were preincubated with the compounds. AMPK activation did not significantly reduce viral replication, indicating that viral entry is inhibited, in accordance with our results involving flaviviruses and dextran. The ULK1 activator reduced rubella virus replication when added prior to entry (1.0 log_10_, *p* = 0.03) but failed to inhibit EV71 replication. This indicates that AMPK induces two distinct antiviral pathways; one is ULK1-dependent and inhibits enveloped viruses, while the other additionally inhibits non-enveloped viruses.

### 3.4. AMPK Activation Does Not Inhibit SARS-CoV-2 Entry into Calu-3 Cells but Does in Vero Cells

Recently, we demonstrated that ASA and SA inhibit SARS-CoV-2 replication in cell culture and human PCLSs [23]. Quantifying viral RNA in infected cells at different time points did not support a correlation between the inhibition of viral replication and the interference with viral entry in Calu-3 cell cultures. However, as SARS-CoV-2 can enter cells via direct fusion with the plasma membrane or through endocytosis using the same receptor [42,43], we reanalysed the influence of SA on entry using a modified experimental approach with Calu-3 or Vero cells (Figure 4A,B). Calu-3 cells express high amounts of the TMPRSS2 protease and, therefore, promote viral entry by fusion at the plasma membrane. Entry into Vero cells is preferentially endocytic, which we reinforced by adding the TMPRSS2 inhibitor nafamostat (Figure 4B). This experiment enabled the determination of AMPK-mediated influences on the entry pathways since the virus and the receptor remain invariant.

The Calu-3 cells were either preincubated for 2 h before infection with 3 mM SA, or they were infected with SARS-CoV-2, after which 3 mM SA was added 2 h post-infection. In the first case, viral replication should be inhibited if SA targets viral entry, whereas in the second scenario, it should not be affected. The medium was exchanged after 6 h, and viral genomes were quantified after 72 h by RT-qPCR (Figure 4). Virus replication in Calu-3 was independent of pre- or post-incubation, underlining that AMPK did not target entry via direct fusion at the plasma membrane and confirming our previous findings.

To further analyse whether AMPK activation through a combination of SA and metformin inhibits SARS-CoV-2 in a patient-near system, we treated human PCLSs with 5 mM metformin and 1.5 mM SA (Figure 4C). The combination effectively suppressed viral replication, demonstrating that AMPK activation inhibits SARS-CoV-2 in a patient-near system.

A similar experiment with Vero cells showed that AMPK activation by ASA or SA before entry inhibited SARS-CoV-2 replication. However, the antiviral effect was completely abolished when viral entry occurred prior to AMPK activation, demonstrating that ASA and SA specifically target endocytic viral entry rather than receptor recognition or binding. Further inhibition of TMPRSS2 by nafamostat slightly decreased viral replication, indicating that TMPRSS2 activity in Vero cells is low. These results suggest that AMPK activation can inhibit SARS-CoV-2 entry in cells expressing low amounts of TMPRSS2 and elucidate the observed correlation between fasting blood glucose and the severity of COVID-19 in patients, even without diagnosed diabetes [44,45]. In addition, AMPK activation initiates a second antiviral mechanism, blocking SARS-CoV-2 replication in Calu-3 cells.

### 3.5. AMPK Activation Inhibits Entry of Negative-Stranded RNA Viruses, Such as Rabies Virus, and Supports Rabies Postexposure Prophylaxis

Since we have demonstrated that the endocytic entry of (+)-stranded RNA viruses is AMPK-dependent, we investigated whether the entry of negative-stranded RNA viruses, specifically the vesicular stomatitis virus and the rabies virus, is influenced by AMPK activation. In particular, if ASA/SA inhibits rabies replication, it could be included in the postexposure prophylaxis alongside antisera to prevent rabies infections. Inhibition and entry analyses were conducted as described above.

BHK-21 or mouse NA 42/13 cells were either pre-treated with 3 mM ASA/SA, 100 µM AMPK, or ULK1 activators and subsequently infected with the rabies vaccine (BHK-21 cells, Figure 5A) or the wild-type strain (NA 42/13 cells, Figure 5B), or they were infected and incubated with the compounds after 2 h. Supernatants were harvested after 75 h, and viral genome copies were determined by RTqPCR. Pre-treatment with SA inhibited replication of the rabies vaccine strain by more than 1.7 log_10_ (SA: 1.71 log_10_, *p* = 0.03). AMPK activation inhibited entry of the vaccine strain by 1.4 log_10_ (*p* = 0.04) and the wild-type rabies virus by approximately 1.5 log_10_ (SA: 1.6 log_10_, *p* = 0.002, AMPK activator: 1.5 log_10_, *p* = 0.002) into NA 42/13 cells, while the ULK1 activator reduced entry of the vaccine strain by 1.1 log_10_ (*p* = 0.046) and the wild-type rabies virus by 0.5 log_10_ (*p* = 0.013). Thus, activation of AMPK downregulates the entry of the rabies virus.

The AMPK-induced inhibition of viral replication was confirmed in human stem cell-derived 3D neuronal organoid models. The organoids were exposed to 1.5 or 3 mM SA and subsequently infected with wild-type rabies virus. The medium was exchanged after 24 h to remove the non-incorporated virus. Supernatants were collected after 3 d of infection, and viral RNAs were quantified by RTqPCR (Figure 5C). The introduction of 3 mM SA reduced viral genome amounts by 1 log_10_ (*p* = 0.05). Similar inhibition of the vesicular stomatitis virus was observed after AMPK activation with SA (Appendix A).

AMPK activation inhibited rabies wild-type and vaccine strain entry and replication. Thus, it might be beneficial for exposed patients to add an AMPK activator to the approved antisera for postexposure prophylaxis or, if antisera are unavailable, use them to clean the bite wounds.

### 3.6. AMPK Activation Does Not Influence Receptor Presentation but Downregulates TXNIP, Rab5, and Rab7 by Inactivating eEF2 and Activating the Proteasome

We further investigated the mechanisms involved in the AMPK-mediated downregulation of viral entry. Viruses, such as DENV, TBEV, YFV, chikungunya, rabies, VSV, and SARS-CoV-2 or molecules, such as dextran, utilise specific entry receptors and should be regulated differently. Consequently, AMPK activation is hypothesised to impede viral entry and, more broadly, receptor-mediated endocytosis. First, we determined if AMPK or ULK1 activation changed the receptor presentation at the plasma membrane. We decided to analyse the poliovirus receptor CD155 and the VSV low-density lipoprotein (LDL) entry receptor on MT4 cells by FACS [46] to address receptor proteins of negative- and non-enveloped, positive-stranded RNA viruses. The LDL receptor supports the entry of Getah, Semliki Forest, and Bebaru viruses in addition to VSV [47]. MT4 cells were treated with the AMPK or ULK1 activator for 2 h and stained with the respective rabbit-anti-CD155 or mouse-anti-LDL receptor antibodies. FACS analyses showed that the amounts of both receptors did not decrease as a result of AMPK or ULK1 activation (Figure 6A and Appendix A), indicating that changes in the receptor concentration are not responsible for the observed entry inhibition.

Generally, the α-arrestin TXNIP is known to regulate the endocytosis of GLUT1. TXNIP has a short half-life of about 12 min [48] and is negatively regulated by AMPK through phosphorylation, which promotes its degradation and causes the inhibition of GLUT1 endocytosis. To investigate whether AMPK-mediated inhibition of viral entry is related to TXNIP degradation. Huh-7 cells were incubated with SA for 24 h or an AMPK activator for 2 h, and TXNIP expression was visualised by Western blotting using specific antibodies (Figure 6B). The multiple TXNIP forms were probably due to degradation. Activation of AMPK decreased TXNIP amounts after 2 h and 24 h, suggesting that one of the first steps of endocytosis is already inhibited, similar to the inhibition of GLUT1. In contrast, ULK1 activation reduced TXNIP only after 24 h.

AMPK activates the eEF2 kinase, resulting in the inhibition of translation. The inhibition of translation combined with the short half-life of TXNIP could explain the protein loss due to AMPK activation. The cells were treated with cycloheximide (50 µg/mL) for 2 h to analyse whether the downregulation of TXNIP was due to the inhibition of translation. TXNIP expression was downregulated by inhibition of translation, similar to SA treatment, suggesting that TXNIP is regulated by eEF2 kinase-mediated inactivation (Figure 6B upper panel). However, ULK1 activation did not significantly decrease TXNIP after 2 h, indicating that the direct phosphorylation by AMPK, which was shown to promote TXNIP degradation by the proteasome [8], and the inhibition of translation is required for TXNIP reduction. We confirmed the results using immunofluorescence microscopy (Figure 6C), where TXNIP was reduced by cycloheximide and AMPK activation but not by ULK1 activators. The nuclear localisation of TXNIP has been reported before [8,49].

Cycloheximide treatment should block the uptake of dextran if the de novo synthesis of TXNIP is essential for entry. Thus, we treated Huh-7 cells with cycloheximide and added fluorescently labelled dextran. The incorporation was visualised with confocal microscopy (Figure 1B). SA treatment abolished dextran uptake, while cycloheximide reduced dextran endocytosis compared to the untreated control, indicating that TXNIP enhances dextran uptake.

The p65 subunit of NF-κB downregulates the expression of the eEF2 kinase [2]. Thus, overexpression of p65 should alleviate the AMPK-mediated inhibition of viral entry if its downregulation is eEF2K-dependent. We constructed a constitutively active p65 S267D mutant expression plasmid. 293T cells were transfected with either the parental vector or the pcDNA-p65 S276D plasmid. The cells were split, treated with SA or ASA, and infected with YFV (Figure 6D) or SARS-CoV-2 (Figure 6E). AMPK activation by ASA or SA reduced YFV replication by over 2 orders of magnitude in control cells (Figure 6D,E). The expression of the constitutively active NF-κB p65 S276D rescues most of the effect of AMPK on viral replication to less than one order of magnitude, indicating that NF-κB restores viral entry. Similar results were obtained for SARS-CoV-2, where AMPK activation inhibited replication by more than 3.6 orders of magnitude. The expression of the constitutively active NF-κB p65 reconstituted viral replication.

In addition to the downregulation of translation, the degradation of TXNIP could contribute to the shutdown of endocytosis through AMPK activation. Therefore, we sought to investigate whether AMPK activation would influence the activity of the proteasome. Huh-7 cells were incubated for 2 h with AMPK and ULK1 activators, cycloheximide, and the proteasome inhibitor bortezomib (100 nM). The activation of AMPK and ULK1 upregulated the cellular chymotrypsin-like protease activity by 5-to-9-fold (Figure 6F), while only ULK1 activation influenced caspase-like protease activity. TXNIP encodes several predicted highly specific chymotrypsin cleavage sites, while caspase cleavage sites are absent. However, it has been shown that TXNIP is downregulated by caspase activation in T cells. [50]. The kinase activators did not affect the activity of trypsin-like proteases. The upregulation of proteasome activity could explain the incomplete reconstitution of viral entry by the constitutively active NF-κB p65.

### 3.7. AMPK Activation Rapidly Downregulates Rab5, Rab7, and TXNIP

The Ras-related protein Rab5 marks the EE and is enriched in EEs, serving as the primary regulator of the conversion to late endosomes (LEs), where Rab5-GTP recruits Rab7 to the endosome. Thus, EEs are marked by Rab5, whereas during the transition from EEs to LEs, Rab7 replaces Rab5. Analyses of Rab5 and Rab7 amounts by Western blotting revealed that both proteins are downregulated by AMPK and, surprisingly, by ULK1 activation (Figure 6B), indicating that ULK1 influences the Rab5-positive EEs. These results were confirmed by confocal microscopy using Rab5- and Rab7-specific antibodies (Figure 7A). Vero cells were treated for 2 h with AMPK or ULK1 activators, then fixed and stained. Confocal microscopy revealed that AMPK activation by SA or the AMPK activator reduced the expression of Rab5 and Rab7. Moreover, ULK1 activation also decreased the number of LEs and EEs. Furthermore, Rab5 and -7 are not only regulated by inhibition of translation but by the activation of the proteasome as well. Thus, the AMPK pathway regulates EEs and LEs, in addition to the Rab4-mediated endosome recycling that has been previously reported [14]. Considering our results, the upregulation of Rab4 might compensate for the downregulation of Rab5 in EEs and Rab7 in LEs.

Next, we aimed to determine whether Rab5 and Rab7 were downregulated, similar to TXNIP, through AMPK-mediated inhibition of translation or activation of the proteasome. Cells were treated with cycloheximide, and Rab5 and Rab7 were visualised using immunofluorescence (Figure 7B). The downregulation of translation by cycloheximide resulted in reduced levels of Rab5 and Rab7, similar to that observed during AMPK activation. This reliance on de novo translation is unexpected, given that the half-life of Rab7 has been determined to be 28 h and may also be attributed to the upregulation of proteasomal proteases. Therefore, we examined the time-dependency of SA-mediated AMPK activation on Rab5, Rab7, and TXNIP (Figure 7C and Appendix A, which includes additional time points). SA decreased Rab5 and Rab7 to undetectable levels after 2 min, while the TXNIP signal remained detectable after 8 min. After 15 min of treatment, neither Rab nor TXNIP signals were observed. However, low levels of Rab7 re-emerged after 2 h. These results substantiate the rapid inhibition of endocytosis and viral entry, indicating that AMPK activation significantly diminishes the half-life of these cellular factors.

The results indicate that AMPK activation inhibits the initial step of endocytosis, as evidenced by the decreased levels of TXNIP in a ULK1-independent way. Nevertheless, both AMPK and ULK1 reduce cellular Rab5 and Rab7 levels within minutes, suggesting a decline in the later stages of endocytosis through translation and proteasome activation (refer to the graphical abstract). In particular, the rapid downregulation of all three endocytosis factors reinforces the observed inhibition of viral entry through drug-induced activation of the AMPK pathway.

### 3.8. AMPK Activation Modifies the Endo-Lysosomal SARS-CoV-2 Replication Compartment

Since Rab7 recruits the V-ATPase required for converting endosomes to lysosomes, we analysed whether AMPK activation inhibits SARS-CoV-2 replication at post-entry steps. First, we determined the influence of AMPK activation on viral gene expression in these cells. Calu-3 cells were incubated with ASA and were infected with SARS-CoV-2. Cells were lysed 3 days after infection, and the amounts of viral N- and spike proteins were detected through Western blotting analysis. Cell culture supernatants were also collected, and the viruses were concentrated by ultracentrifugation through a sucrose cushion. Treatment with AMPK activation reduced the non-processed S protein below the detection level (Figure 8A), while S and S2 were absent in the supernatant. However, detectable amounts of the N proteins were found in the concentrated viruses.

SARS-CoV-2 replication extensively utilises the lysosomal compartment, and AMPK also regulates lysosomal function as well as the endosomal–lysosomal compartment. Recently, we demonstrated that the serotonin reuptake inhibitor fluoxetine traps SARS-CoV-2 in the lysosomes [51]. Here, we sought to analyse whether AMPK activation similarly modifies the replication compartment. Calu-3 cells were incubated with ASA and infected with SARS-CoV-2 at an MOI of 10. After 24 h, the cells were lysed, and the lysosomes were isolated. The lysosomal pellets were dissolved in PBS, and viral RNAs and marker proteins were analysed. We determined a reduction of more than 3 log_10_ of viral RNAs, with S protein levels being almost undetectable in the cells treated with ASA compared to those in the untreated control (Figure 8B,C), indicating that AMPK activation inhibited viral transcription and genome replication, resulting in decreased S and N protein levels. Analyses of the endoplasmic reticulum marker GRP78 revealed that the expression of this protein, similar to that of the lysosomal marker protein LAMP1, was reduced in our lysosomal fraction (Figure 8D,E). These results indicate that AMPK activation with ASA leads to fewer replicative lysosomal compartments, consistent with the observed reductions in Rab5 and Rab7 levels. Our findings suggest that ASA treatment modifies endosomal, lysosomal, and SARS-CoV-2 replication compartments.

### 3.9. The PI3K/Akt Kinase Pathway Is Essential for Flavivirus and SARS-CoV-2 Replication

The PI3K/Akt pathway promotes viral endocytosis by activating mTORC1 and silencing AMPK/ULK1. The activation of AMPK antagonises the PI3K/Akt pathway by inhibiting raptor. Thus, we sought to investigate whether the downregulation of the PI3K/Akt pathway influences viral replication by chemically inhibiting several pathway kinases. Huh-7 cells were preincubated with the PI3K inhibitor LY294002 (40 µM) or 300 nM of the novel clinically approved inhibitor PI3K inhibitor copanlisib and were subsequently infected with YFV. Cell culture supernatants were collected 48 h after infection, and viral RNAs were quantified by RTqPCR (Figure 9A). Western blotting analyses of the treated cells revealed that the PI3K inhibitors reduced Akt serine 473 phosphorylation, while AMPK amounts remained unchanged (Figure 9B). Inhibition of the PI3K through preincubation with LY294002 reduced viral replication by 2.44 log_10_ (*p* < 0.0001) and by 1.9 log_10_ with copanlisib (*p* = 0.0001). This indicated that the regulation of AMPK by the activity of the PI3K/Akt signalling cascade is crucial for viral replication (Figure 9A). Similarly, treatment of Calu-3 cells with LY294002 inhibited SARS-CoV-2 replication by 2.63 log_10_ (*p* = 0.002), supporting the importance of PI3K activity (Figure 9B).

The PI3K is upstream of the Akt kinase, which controls the activation of NF-κB and, in turn, the expression of TXNIP. Thus, we sought to investigate whether the activity of Akt influences SARS-CoV-2 and flavivirus replication. The Akt kinase facilitates the phosphorylation of mTOR. The Akt-I/II inhibitor blocks the activity of Akt kinase I and II and, thus, autophosphorylation. Huh-7 and Calu-3 cells were incubated with 58 µM of Akt-I/II inhibitor and infected with YFV or SARS-CoV-2. Western blotting analyses confirmed reduced Akt phosphorylation (Figure 9B). Viral replication was determined at 48 h (YFV) or 72 h (SARS-CoV-2) after infection by RT-qPCR (Figure 9A,C). Treatment with the Akt-I/II inhibitor downregulated viral replication by more than 3 log_10_ (YFV: 3.6 log_10_, *p* < 0.00001; SARS-CoV-2: 3.84 log_10_, *p* = 0.002). This result indicates that the activity of the Akt kinase is a crucial host factor for viral replication by inhibiting AMPK activation.

If AMPK activation and PI3K/Akt inhibition are two sides of the same coin, the suppression of PI3K or Akt activity should also inhibit viral entry. Thus, we performed an entry assay similar to the experiments above with a PI3K inhibitor. Huh-7 cells were preincubated with LY294002 inhibitor for 2 h and infected or infected and treated with inhibitor 2 h after infection. Viral RNAs were isolated after 48 h and quantified by RTqPCR (Figure 8D). Inhibition of PI3K before infection resulted in the suppression of flavivirus replication by 1.92 log_10_ (*p* < 0.0001), indicating that PI3K activity and the PI3K/Akt pathway are essential for flavivirus replication. However, treatment after infection was ineffective (0.5 log_10_, *p* = 0.009), suggesting that PI3K inhibition suppressed viral entry via increased AMPK activity. The importance of the Akt kinase activity was confirmed in the TBEV and chikungunya entry assays (Figure 2D and Figure 3), where preincubation with the AKT I/II inhibitor suppressed viral replication by 3.3 log_10_ (TBEV, *p* = 0.02) and 4.2 log_10_ (chikungunya, *p* = 1.3 × 10^−5^).

Next, we sought to analyse the downstream effectors of Akt. The activity of the mTOR kinase controls major cellular pathways, such as NF-κB activation [10], cell metabolism, transcription, protein synthesis, autophagy, and the balance of proteasome activity. Furthermore, mTOR regulates the activity of the lysosomal ATPase, which controls the lysosomal pH [52]. The mTOR protein exists in two distinct complexes, mTORC1 and mTORC2. The Akt kinase controls the former, while the latter regulates the Akt kinase independently of PI3K activity. First, we investigated the effects of inhibiting mTOR activity with torin-1 and temsirolimus, a novel clinically approved mTOR inhibitor. NCE cells were either preincubated for 2 h or TBEV-infected and treated after 2 h with 10 µM temsirolimus, while Calu-3 cells were treated with 100 nM torin-1 and subsequently infected with SARS-CoV-2. Viral replication was determined using RT-qPCR at 48 h (TBEV) or 72 h (SARS-CoV-2) (Figure 2D and Figure 9C) post-infection. Inhibition of both mTORC1 and -2 reduced SARS-CoV-2 replication by 1.4 log_10_ (*p* = 0.003) (Figure 8). This demonstrates that mTOR activity is required for viral replication (Figure 8). Furthermore, the TBEV entry experiments confirmed the inhibition of viral entry (1.4 log_10_, *p* = 0.026). Experiments with rapamycin were performed to characterise further the mTORC complex, which inhibits SARS-CoV-2. Rapamycin selectively targets mTORC1. Incubation of Calu-3 cells with rapamycin inhibited viral replication (SARS-CoV-2: 0.9 log_10_; *p* = 0.005), providing evidence that mTORC1 activity is essential for viral replication (Figure 9C). The connection between mTORC1 and flavivirus, as well as SARS-CoV-2 replication, can be explained by control of AMPK activity and, additionally, in the case of SARS-CoV-2, by the mTOR-mediated acidification of the lysosomes.

Next, we analysed whether mTORC2 is required for SARS-CoV-2 and YFV replication, as mTORC2 activates Akt via serine 473 phosphorylation. Calu-3 and Huh-7 cells were incubated with the specific mTORC2 inhibitor JR-AB2-011 (Figure 9A,C). The cells were treated with JR-AB2-011 (250 µM) inhibitor and infected with YFV or SARS-CoV-2. Viral genome copies in the supernatants were quantified 48 h and 72 h post-infection by RT-qPCR. JR-AB2-011 suppressed YFV and SARS-CoV-2 replication by approximately 3 log_10_ (YFV: 2.84 log_10_, *p* = 0.0003; SARS-CoV-2: 3.21 log_10_, *p* = 0.002), indicating that mTORC2 plays a significant role in YFV and SARS-CoV-2 replication, which can lead to Akt activation. Moreover, the activation of AMPK by ASA downregulated the phosphorylation of Akt at serine 473 in either non-infected or SARS-CoV-2-infected Calu-3 cells, mediated by the mTORC2 complex. This provides evidence that AMPK controls not only mTORC1 through phosphorylation of raptor but also mTORC2. This results in the inhibition of viral replication, as we have demonstrated that Akt is essential for SARS-CoV-2 replication.

mTORC1 regulates the translation of specific heat-shock-related mRNAs by controlling the phosphorylation of the ribosomal S6 protein and the translation of 5′TOP-RNAs. Thus, we investigated whether the PI3K signalling cascade inhibits viral replication by downregulating the translation of viral RNA. The S6-kinase inhibitor PF-4708671 specifically suppresses S6 protein phosphorylation. Calu-3 and Huh-7 cells were incubated with 10 µM PF-4708671 and infected with YFV and SARS-CoV-2. Viral RNAs were isolated 48 h and 72 h after infection and quantified by RTqPCR (Figure 9A,C). PF-4708671 did not influence viral replication, showing that S6 phosphorylation is dispensable for flavivirus or SARS-CoV-2 replication (SARS-CoV-2: −0.13 log_10_, *p* = 0.17). However, AMPK activation by ASA downregulated S6 phosphorylation in Calu-3 and Huh-7 cells, which is expected as ASA leads to the dephosphorylation of Akt. We have shown that the S6 phosphorylation is not essential for SARS-CoV-2 replication. Thus, the ASA-mediated inhibition of S6 activity should not be responsible for the antiviral effect of ASA.

In summary, we have shown that active PI3K/Akt and mTORC2 pathways and the repression of AMPK/ULK1 are crucial for flavivirus, alphavirus, enterovirus, SARS-CoV-2, and rabies lyssavirus entry and replication. This inhibition of receptor-mediated endocytosis and viral entry was independent of the AMPK activation pathway since ASA/SA, metformin, and the AMPK activator act on different cellular targets. In addition, our results suggest that the regulation of viral endocytosis is ULK1-dependent. Furthermore, the observed downregulation of EEs and LEs supports the reported AMPK-dependent upregulation of Rab4, an endosome recycling factor [14]. The observed downregulation of LAMP-1 indicates that inhibiting EEs and LEs leads to reduced cellular lysosomes. Our findings suggest that AMPK/ULK1 activation inhibits receptor-mediated endocytosis and changes the number or composition of lysosomes. This central mechanism is a prime target for developing broad-spectrum antivirals. Finally, inhibiting rabies entry might lead to modification of the current postexposure prophylaxis, especially in the absence of antiserum with inexpensive and globally available drugs, such as ASA or metformin.

## 4. Conclusions

In summary, we have demonstrated that active PI3K/Akt and mTORC2 pathways, as well as the repression of AMPK/ULK1, are crucial for the entry and replication of flavivirus, alphavirus, enteroviruses, SARS-CoV-2, and rabies lyssavirus. This inhibition of receptor-mediated endocytosis and viral entry is independent of the AMPK activation pathway (graphical abstract), as ASA/SA, metformin, and the AMPK activator act on distinct cellular targets. We show that AMPK downregulates TXNIP, EEs, Les, and lysosomes via two distinct pathways.

The observed downregulation of Rab5, Rab7, and TXNIP occurs within minutes after drug-induced AMPK activation, similar to the observed antiviral effects. We found evidence that the resynthesis of TXNIP, Rab5, and Rab7 is essential for endocytosis. The translation elongation factor eEF1 is a known target of AMPK (graphical abstract), which inhibits ribosome translocation. NF-κB p65 downregulates eEF2K expression and significantly neutralises the AMPK-mediated downregulation of viral endocytosis. Furthermore, the suppression of translation by cycloheximide resulted in a weaker downregulation of endocytosis factors and a lower inhibition of dextran uptake compared to AMPK activation.

Our results suggest that the regulation of viral endocytosis by AMPK is also ULK1-dependent and upregulates the cellular proteasome. In contrast to AMPK, ULK1 was shown to repress Rab5 and Rab7, not TXNIP, after 2 h. Thus, the suppression of endocytosis is likely mediated by the inhibited resynthesis and the activated degradation of TXNIP and the Rab proteins.

The observed downregulation of LAMP-1 indicates that the inhibition of EEs and LEs leads to reduced cellular lysosomes. Our findings suggest that AMPK/ULK1 activation inhibits receptor-mediated endocytosis and alters the number or composition of lysosomes. This central mechanism represents a prime target for developing broad-spectrum antivirals. Finally, inhibiting rabies entry may lead to modifications in the current postexposure prophylaxis, especially in the absence of antiserum, utilising inexpensive and globally available drugs, such as ASA or metformin.

On the other hand, we have shown that AMPK activation leads to the inhibition of receptor recycling. This could be used to enhance receptor presentation or signalling during immune therapies.

## Figures and Tables

**Figure 1 cells-14-00334-f001:**
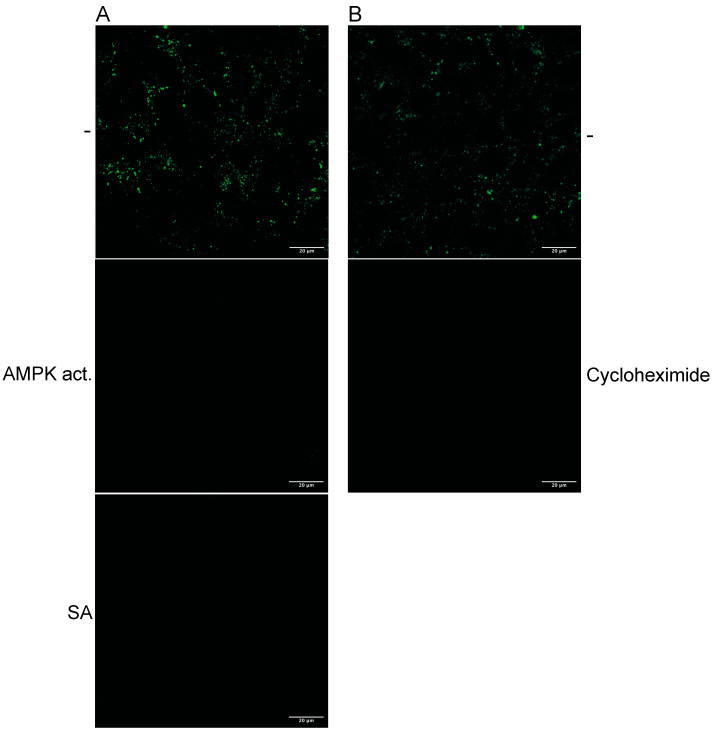
Confocal microscopy of cells incubated with (**A**) 3 mM SA or 100 µM AMPK activator and (**B**) cycloheximide and 2.5 mg/mL fluorescein isothiocyanate–dextran. The cells were fixed after 2 h of treatment.

**Figure 2 cells-14-00334-f002:**
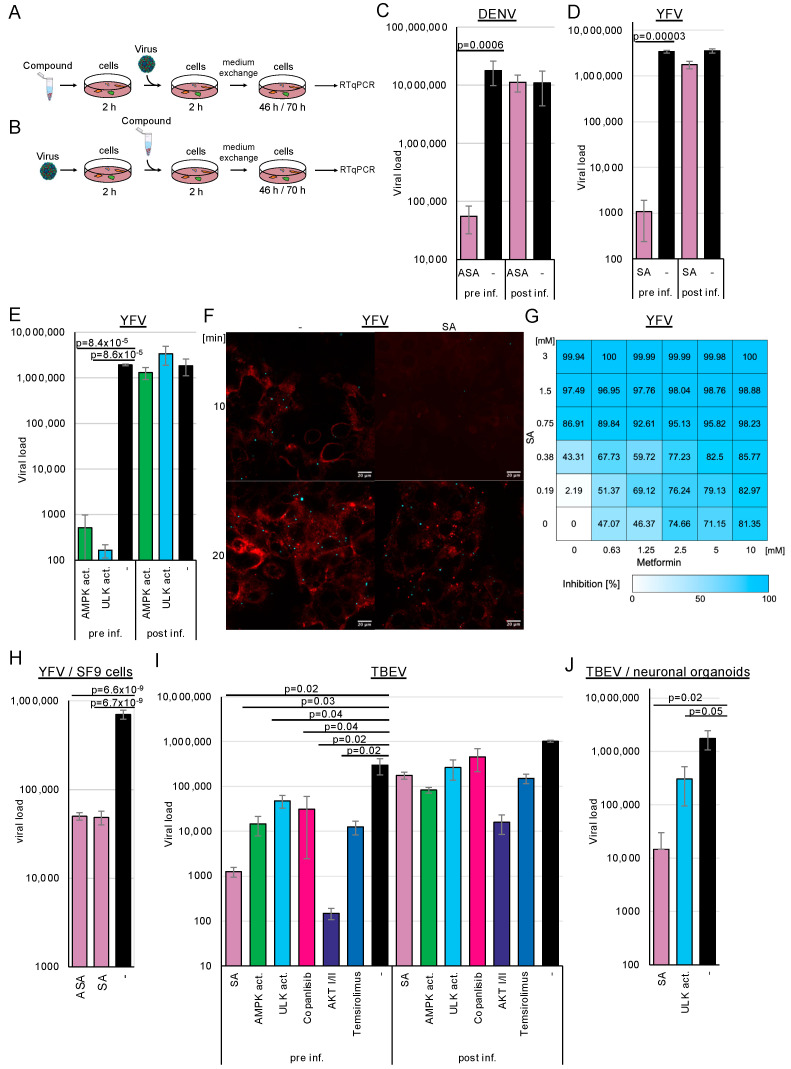
AMPK and ULK1 activation or PI3 kinase inhibition reduces flavivirus entry. (**A**) Cells were pre-treated with the AMPK and ULK1 activators or kinase inhibitors for 2 h and infected with (pre inf.) (**C**) DENV, (**D**,**E**) YFV, or (**I**) TBEV, or (**B**) infected for 2 h and treated (post inf.). (**F**) Confocal microscopy (viruses, cyan; cellular membranes, red) of SA treatment and YFV infection. Pre-treated Huh-7 cells were infected with fluorescently labelled YFV for 10 or 20 min. Scale bar: 20 µm. (**G**) Inhibition of YFV replication by combinations of SA and metformin. (**H**) AMPK activation blocks YFV replication in SF9 insect cells. SA inhibits YFV in insect cells. (**I**) TBEV entry is sensitive to AMPK activation, PI3K inhibitors (copanlisib), AKT kinase inhibitors (AKT I/II), and mTOR inhibitors (temsirolimus). Viral replication was determined using RTqPCR. (**J**) SA and ULK1 activators suppress TBEV replication in neuronal organoids.

**Figure 3 cells-14-00334-f003:**
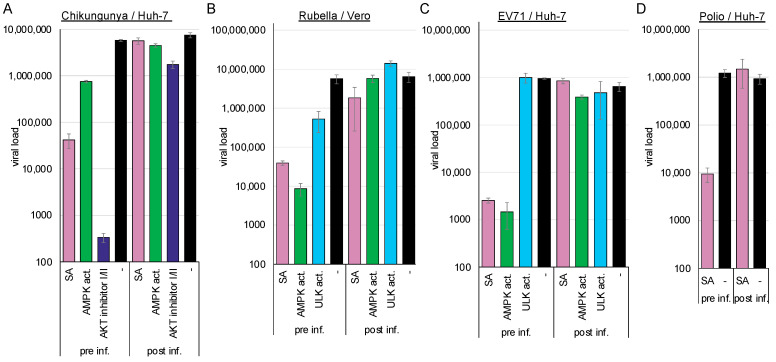
AMPK activation inhibits (**A**) chikungunya, (**B**) rubella, (**C**) EV71, and (**D**) poliovirus entry. Cells were preincubated for 2 h with the compounds and infected with chikungunya (pre inf.) or infected and supplemented with the compounds 2 h after infection (post inf.). Released viral genomes were quantified 48 h after infection. Cell lines and viruses are depicted above the panel.

**Figure 4 cells-14-00334-f004:**
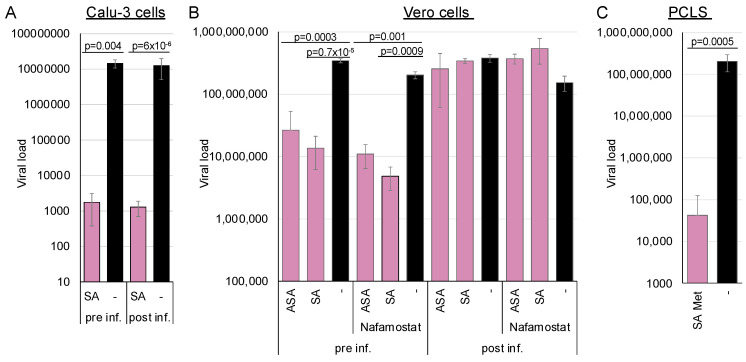
AMPK activation inhibits SARS-CoV-2 in Calu-3 and Huh-7 cells but does not influence viral entry in Calu-3 cells. (**A**) Calu-3 cells or (**B**) Vero cells were preincubated for 2 h with 3 mM SA and infected (pre inf.) or infected and supplemented with 3 mM SA 2 h after infection (post inf.). Nafamostat was added to ensure entry via endocytosis. (**C**) Treatment with 1.5 mM and 5 mM metformin inhibits SARS-CoV-2 in precision-cut lung slices. Released viral genomes were quantified 72 h after infection.

**Figure 5 cells-14-00334-f005:**
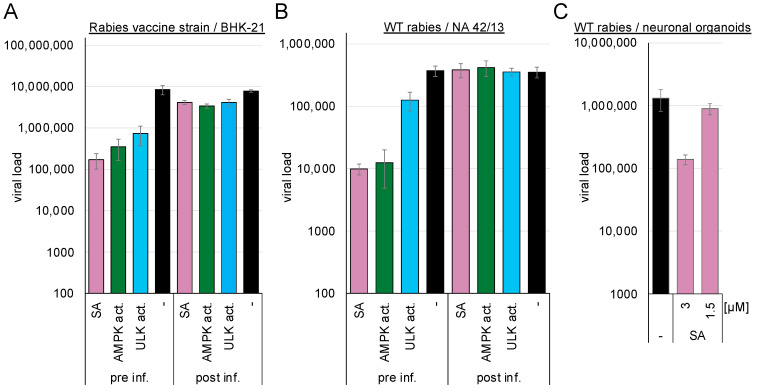
AMPK and ULK1 activation inhibits rabies vaccine and wild-type strains. (**A**) BHK21, (**B**) NA42/13 cells, or (**C**) neuronal organoids were preincubated for 2 h with the compounds (3 mM ASA/SA, 100 µM AMPK, or UKL1 activator) and infected with (**A**) rabies vaccine or (**B**,**C**) wild-type strain (pre inf.) or infected and supplemented with compounds 2 h after infection (post inf.).

**Figure 6 cells-14-00334-f006:**
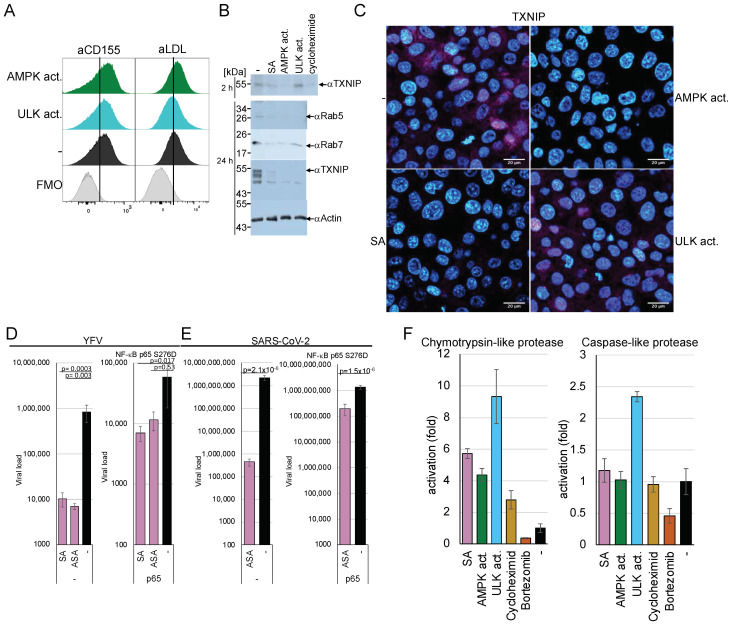
AMPK activation does not influence the surface expression of CD155 or the LDL receptor, but it reduces cellular TXNIP, Rab5, and Rab7 and inhibits viral endocytosis. (**A**) FACS analyses of the CD155 and LDLR expression at the plasma membrane. FMO: control without the first antibodies. (**B**) Western blotting analysis of TXNIP, Rab5, and Rab7 after 24 h and for TXNIP after 2 h. (**C**) Confocal microscopy of TXNIP expression after 2 h. blue: DAPI; magenta: TXNIP. Scale bars 20 µm. (**D**) YFV and (**E**) SARS-CoV-2 replication in 293T cells treated with ASA SA and transfected with a constitutive active NF-κB. (**F**) Analyses of proteasomal proteases with the compounds. Bortezomib served as inhibitor control.

**Figure 7 cells-14-00334-f007:**
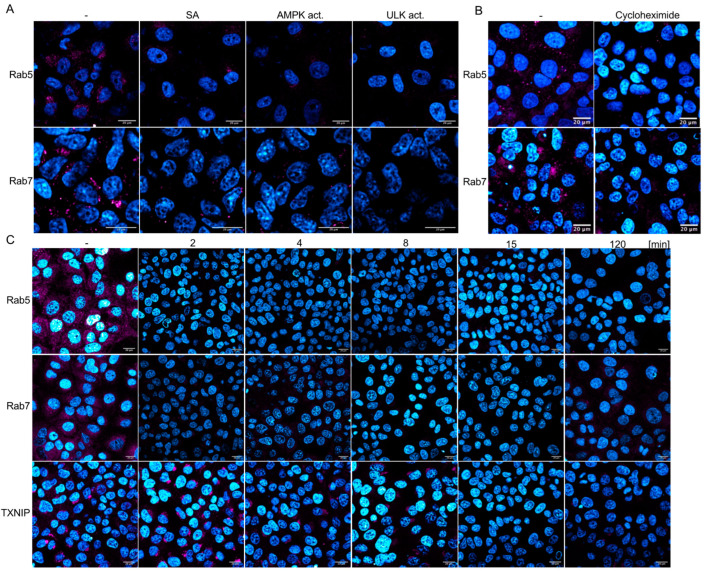
AMPK activation downregulates Rab5, Rab7, and TXNIP rapidly. Confocal microscopy of Rab5 (magenta), Rab7 (magenta), and TXNIP (magenta) after SA (**A**,**C**), AMPK (**A**), ULK activator (**A**), and cycloheximide (**B**) treatment. (**C**) Analyses of the time-dependent downregulation of Rab5, Rab7, and TXNIP. Scale bars: 20 µm.

**Figure 8 cells-14-00334-f008:**
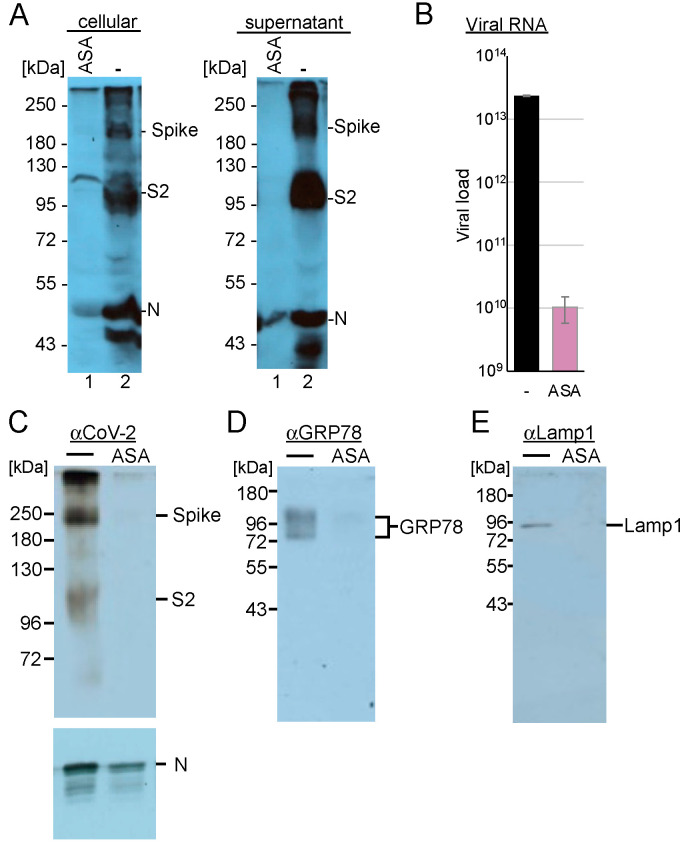
AMPK activation reduces (**A**) spike, nucleocapsid protein (**N**) expression, and lysosomal amounts of (**B**) viral RNA, (**C**) spike protein, (**D**) endoplasmic marker GRP78, and the lysosomal marker protein Lamp1. (**A**) Calu-3 cells were incubated with ASA and infected with SARS-CoV-2 for 72 h. Infected cells and viral supernatants were analysed with anti-SARS-CoV-2 S2 and anti-SARS-CoV-2 N protein antibodies. (**B**) RTqPCR quantification and (**C**–**E**) Western blotting analyses of the lysosomal fraction of SARS-CoV-2-infected Calu-3 cells.

**Figure 9 cells-14-00334-f009:**
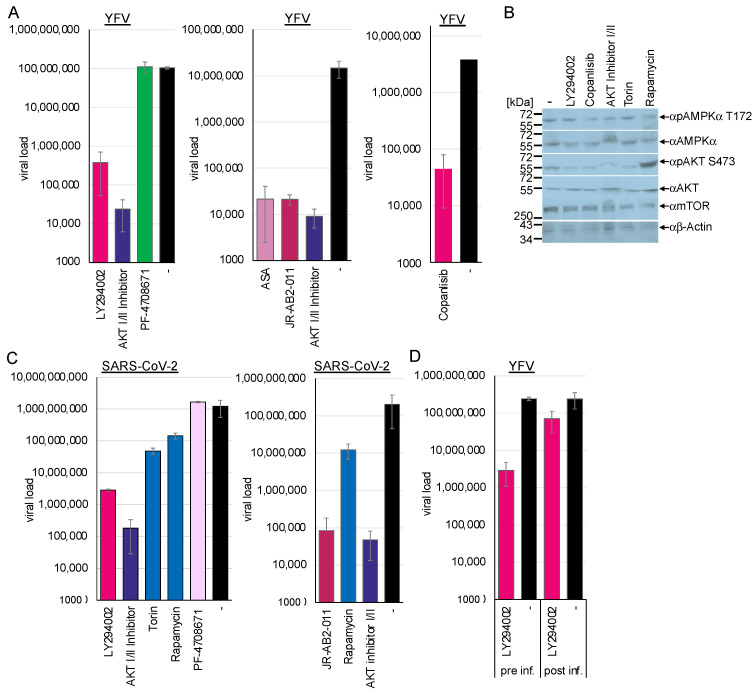
The PI3K, mTORC2, Akt, and mTORC1 are essential host factors for (**A**) flavivirus and (**C**) SARS-CoV-2 replication—(**D**) PI3K activity is required for YFV entry. (**A**,**D**) Huh-7 cells or (**C**) Calu-3 cells were incubated with inhibitors of PI3K (LY294002, copanlisib), Akt (AKT I/II), TORC1/2 (Torin), TORC1 (rapamycin), S6 kinase (PF-4708671), and TORC2 (JR-AB2-011) and infected with (**A**,**D**) YFV or (**C**) SARS-CoV-2. Viral replication was analysed 72 h after infection. (**B**) Influences on protein expression and modifications were determined by Western blotting (**D**) YFV entry assays.

## Data Availability

The original contributions presented in this study are included in the article/Appendix A. Further inquiries can be directed to the corresponding author.

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
