# Peer review of "AMPK Activation Downregulates TXNIP, Rab5, and Rab7 Within Minutes, Thereby Inhibiting the Endocytosis-Mediated Entry of Human Pathogenic Viruses"

_cells, 2025, doi:10.3390/cells14050334_

Round 1
Reviewer 1 Report (Previous Reviewer 2)
Comments and Suggestions for Authors
AMPK/ULK1 activation downregulates TXNIP, Rab5, and Rab7 and inhibits endocytosis-mediated entry of human pathogenic viruses
The authors have included additional mechanistic data in the revised manuscript, which is appreciated.
Major comments
1. Figure 6C: What is the subcellular localization of TXNIP? Does it reside primarily in the nucleus or cytoplasm? Support your response with relevant evidence and references.
2. What is the conclusion of the paragraph spanning lines 488–499? How is TXNIP degraded, and what type of degradation mechanism is involved?

Author Response
Please see the attachment

Reviewer 2 Report (Previous Reviewer 1)
Comments and Suggestions for Authors
Major
I propose conducting further experiments to address important areas that could greatly reinforce the manuscript's findings. The study should start by applying RNA interference or CRISPR-mediated knockdown strategies to AMPK and ULK1 to verify their specific functions in blocking viral entry and replication. Use live-cell imaging to observe the real-time dynamic localization shifts and trafficking patterns of TXNIP, Rab5, and Rab7 when AMPK activation occurs. Conduct detailed dose-response assessments for ASA, metformin, and AMPK activators in various cell types including patient-derived models to determine dosage-dependent effectiveness and wider applicability. The localization of Rab5 and Rab7 should be evaluated in different conditions to show the impact of treatment in functionality of these proteins.
Minor:
1- The quality of Confocal microscope image should be improved.
2- Please expand the future direction.
Round 2
Reviewer 1 Report (Previous Reviewer 2)
Comments and Suggestions for Authors
The reviewer’s questions have been addressed in the revision, and they are satisfied with the responses.
Comments on the Quality of English LanguageThis reviewer does not see major issues in the manuscript language.
Reviewer 2 Report (Previous Reviewer 1)
Comments and Suggestions for Authors
The authors improved the quality of paper and have done their best to improve the paper. I appreciate it.
This manuscript is a resubmission of an earlier submission. The following is a list of the peer review reports and author responses from that submission.
Round 1
Reviewer 1 Report
Comments and Suggestions for Authors
1) Synopsis
The study investigates the effect of AMPK and ULK1 activation on receptor-mediated endocytosis and the replication of various human pathogenic RNA viruses, such as SARS-CoV-2 and flaviviruses. The authors demonstrate that activation of the AMPK/ULK1 pathway inhibits early and late endocytic steps essential for viral entry, primarily by modulating proteins such as TXNIP, Rab5, and Rab7. The findings suggest potential antiviral implications of AMPK/ULK1 pathway modulation.
2) Strengths
- Innovative Therapeutic Potential: The study offers insights into targeting host pathways, specifically AMPK/ULK1, as broad-spectrum antiviral strategies, which could bypass issues related to viral mutation and drug resistance.
- Extensive Experimental Approach: Multiple techniques, including confocal microscopy, Western blotting, RT-qPCR, and in vitro infection assays, provide robust data across various cell models and viral strains.
- Clinical Relevance: The findings have potential clinical applications, as some agents activating the AMPK pathway, such as aspirin, are widely accessible and could serve as adjunct therapies in viral infections.
3) Weaknesses
- Descriptive Nature with Limited Mechanistic Insight: While the study shows AMPK/ULK1 pathway activation impacts viral entry and replication, it does not elucidate detailed mechanisms linking these pathways to viral inhibition.
- Lack of Autophagic Pathway Analysis: The study fails to explore the role of macroautophagy or autophagic flux changes resulting from AMPK/ULK1 activation, especially in the context of the PI3K/AKT pathway.
- Broad Focus on Multiple Viruses: The study spans a variety of viral models, but lacks depth in analyzing viral-specific pathways and responses to AMPK/ULK1 activation, which could limit the specificity and generalizability of the findings.
4) Suggestions for Major Revisions and Experimental Improvements
Mechanistic Exploration of AMPK/ULK1-Induced Viral Inhibition
- Macroautophagy Involvement: Given the role of AMPK and ULK1 in autophagy, it is crucial to assess how autophagy induction affects viral replication. The authors should:
- Measure LC3B-II and p62 accumulation via Western blot or confocal microscopy to determine autophagic flux, both with and without AMPK/ULK1 activators.
- Use an autophagy inhibitor (e.g., chloroquine or bafilomycin A1) to discern whether the antiviral effects are autophagy-dependent, comparing viral replication with and without autophagy suppression.
Experimental Design to Link PI3K/AKT Pathway to Viral Entry Inhibition
- Since the PI3K/AKT pathway modulates autophagy and endocytosis, further examination is warranted. Suggested experiments:
- Employ PI3K and AKT inhibitors (e.g., LY294002 or MK-2206) to observe changes in viral entry and replication under conditions of pathway suppression, determining if viral inhibition remains consistent.
- Perform calcium mobilization assays using Fluo-4 AM or similar indicators to test if PI3K/AKT modulation affects calcium signaling. This is essential to link endocytosis inhibition and intracellular calcium as possible modulators of viral entry.
Role of Mitochondrial Function in AMPK/ULK1-Mediated Viral Inhibition
- As AMPK regulates mitochondrial dynamics and calcium mobilization, connecting mitochondrial health with viral inhibition would be insightful. Suggested steps:
- Measure mitochondrial membrane potential (Δψm) using JC-1 or TMRE dyes after AMPK/ULK1 activation. This could reveal if mitochondrial dysfunction correlates with decreased viral entry or replication.
- Use mitochondrial calcium imaging with Rhod-2 AM to assess if AMPK/ULK1 activation alters mitochondrial calcium levels, further affecting apoptosis and viral replication.
Investigate TXNIP and Rab Protein Pathways in Viral Entry
- The involvement of TXNIP and Rab GTPases in endocytosis and viral entry warrants additional depth:
- Conduct co-immunoprecipitation assays to evaluate interactions between TXNIP/Rab proteins and viral proteins upon AMPK/ULK1 activation.
- Perform Rab5 and Rab7 knockdowns using siRNA or CRISPR to assess their individual roles in viral entry inhibition following AMPK/ULK1 activation.
5) Comments to Authors
The manuscript provides a detailed analysis of AMPK/ULK1 activation and its antiviral effects, yet remains primarily descriptive and lacks significant mechanistic depth. To improve the impact and clinical relevance, we recommend exploring the influence of PI3K/AKT and macroautophagy on viral replication, as these pathways likely intersect with AMPK/ULK1 activation. A focus on autophagic flux and mitochondrial health could offer novel insights, enabling a clearer understanding of AMPK/ULK1-mediated viral inhibition.
Author Response
Please see the attachment
Kind regards
Jochen Bodem

Reviewer 2 Report
Comments and Suggestions for Authors
AMPK/ULK1 activation downregulates TXNIP, Rab5, and Rab7 and inhibits endocytosis-mediated entry of human pathogenic viruses
The authors of this study identified AMPK and ULK activation suppresses virus replication inhibiting endocytosis. Further they identified that early and late SARS Cov2 replication is impaired by AMPK activation, reducing early endocytic factors TXNIP, Rab5 and the late endosomal marker Rab7 and lysosomal markers LAMP1 and GRP78. The manuscript includes many phenotypic data including enveloped and non-enveloped virus replication data upon the pre and post treatment of AMPK and ULK activators. Mechanistic data regarding how AMPK and ULK activation inhibits endocytic and lysosomal factors are not presented.
There are publications demonstrating that AKT pathway inhibitor treatment impairs the virus progeny regarding SARS CoV2 and YFV. It is obvious that AMPK is inhibited by the activated AKT/PI3K. Thus, using AKT inhibitor to show the activation of AMPK which reduces virus replication is obvious. Authors have used PI3K-AKT-AMPK-mTORC pathway inhibitors to show that AMPK activation by upstream AKT and PI3K molecular inhibitors and inhibitors for AMPK activating molecules of downstream which are expected.
Major comments
1. Additional mechanism data is essential. Especially in the Figure 6B, Figure 7A and C, authors must determine AMPK and mTORCs expression level and their phosphorylation levels in western blots. For the Figure 8, authors must show AKT, AMPK and mTORCs levels and their phosphorylation levels using western blots.
2. Multiple western blot bands are showed in TXNIP blot in the Figure 6B. Please explain the event. Please provide catalog number of each antibody used in this study in the materials and method section.
3. Do authors believe TXNIP is degraded by AMPK activation or does AMPK activation reduced the TXNIP translation (see line 392).
4. The explanation for the copanlisib, AKT I/II and Temsirolismus in the Figure 2D is missing. They should be explained the first time they are referenced in the manuscript. What is Temsirolismus used in this study. Manuscript does not mention about Temsirolismus.
5. Why authors did not use ULK activators to check the DENV (figure 4 and 8), chikungunya, rubella, EV71, polio, SARS Cov2, and Rabies vaccine strain replication. Is there a specific reason? Authors must show ULK activator treatment and virus replication according to the title of this manuscript.
6. Please provide relevant citations for line 463-465.
7. Figure S2 is not cited anywhere in the manuscript.
8. Use exact terms or synonyms for the AMPK and ULK1 activators used in the manuscript when figures are arranged.
9. Figure 2D, Figure 3, Figure 5, Figure 7B, and Figure 8 – Display p values for the data of post infection
10. Supplementary figures are arranged poorly. Revise the figures accordingly.
11. Please recheck the labelling of confocal figures and revise them accordingly.
12. Figure 6 – Revise the figure caption
This reviewer recommends an English editing service for this manuscript.
Author Response
Dear reviewer,
Please see the attachment.
Thank you for being so supportive!
Kind regards,
Jochen Bodem

Reviewer 3 Report
Comments and Suggestions for Authors
The authors demonstrated that activation of AMPK or the downstream ULK1 inhibits receptor-mediated endocytosis. This activation prevents endocytic uptake of human pathogenic enveloped and non-enveloped, positive and negative-stranded RNA viruses, such as yellow fever, dengue, tick-borne encephalitis, chikungunya, polio, rubella, rabies lyssavirus and SARS-CoV-2 not only in mammalian and insect cells but in precision-cut lung slices and neuronal organoids. They suggested that AMPK and ULK1 emerge as restriction factors of cellular endocytosis, impeding the receptor-mediated endocytic entry of enveloped and non-enveloped RNA viruses.
In general, the conclusions drawn by the authors are supported by the data provided. The modification suggestions are as follows:
1. What's the name of AMPK activator? The name of the inhibitor should be used instead of “AMPK act”or “ULK1 act” in text and figures.
2. Figure 1B should be drawn alone, not together with 1A.
3. If Figure1A is the author's summary of the results of this study, it should be the last figure in this article.
4. How to determine the concentration of different inhibitors? The results are missing.
5. The line 555-568 is the same as 569-583.
Author Response
Dear Reviewer,
Please see the attachment.
Kind regards,
Jochen Bodem

Round 2
Reviewer 1 Report
Comments and Suggestions for Authors
I am not convinced with your response about autophagy. p62 is just begining of autophagy investigation. Lack of change of p62 only shows lack of degredation of autophagosome. LC3 lipidation and then using Bafilomycin or CQ to confirm the findings are mandatory.
Comments on the Quality of English LanguageDear Editor,
The quality of English is fine.
With Best Regards
Saeid Ghavami, PhD
Reviewer 2 Report
Comments and Suggestions for Authors
AMPK/ULK1 activation downregulates TXNIP, Rab5, and Rab7
and inhibits endocytosis-mediated entry of human pathogenic
viruses
The authors' response to the revision is unsatisfactory. Consequently, this reviewer is reiterating similar questions that were not adequately addressed. Please carefully review the revision, thoroughly understand the questions, and provide clear and concise answers.
Major comments
1. Please include the western blot data in the manuscript as requested in comment 1 from the first revision.
2. In Figure 6B, multiple bands are observed in the TXNIP western blot. Please clarify the reasons for this occurrence. Give an explanation.
3. Do the authors believe that TXNIP is degraded due to AMPK activation, or does AMPK activation reduce TXNIP translation (refer to line 490 in the revise manuscript)? Give an explanation. Kindly conduct the necessary experiments to provide confirmation.